# Comparison of Supervised and Unsupervised Approaches for the Generation of Synthetic CT from Cone-Beam CT

**DOI:** 10.3390/diagnostics11081435

**Published:** 2021-08-09

**Authors:** Matteo Rossi, Pietro Cerveri

**Affiliations:** Department of Electronics, Information and Bioengineering, Politecnico di Milano, 20133 Milan, Italy

**Keywords:** image-to-image translation, synthetic images, supervised training, unsupervised training, U-Net, cycleGAN, CBCT, CT

## Abstract

Due to major artifacts and uncalibrated Hounsfield units (HU), cone-beam computed tomography (CBCT) cannot be used readily for diagnostics and therapy planning purposes. This study addresses image-to-image translation by convolutional neural networks (CNNs) to convert CBCT to CT-like scans, comparing supervised to unsupervised training techniques, exploiting a pelvic CT/CBCT publicly available dataset. Interestingly, quantitative results were in favor of supervised against unsupervised approach showing improvements in the HU accuracy (62% vs. 50%), structural similarity index (2.5% vs. 1.1%) and peak signal-to-noise ratio (15% vs. 8%). Qualitative results conversely showcased higher anatomical artifacts in the synthetic CBCT generated by the supervised techniques. This was motivated by the higher sensitivity of the supervised training technique to the pixel-wise correspondence contained in the loss function. The unsupervised technique does not require correspondence and mitigates this drawback as it combines adversarial, cycle consistency, and identity loss functions. Overall, two main impacts qualify the paper: (a) the feasibility of CNN to generate accurate synthetic CT from CBCT images, which is fast and easy to use compared to traditional techniques applied in clinics; (b) the proposal of guidelines to drive the selection of the better training technique, which can be shifted to more general image-to-image translation.

## 1. Introduction

Over the past few years, evidence has been accumulating in support of the view that machine learning and artificial intelligence methods may sensibly impact diagnostic and therapeutic areas. In this regard, clinical tools have been recently proposed in oncology to speeding up lesion analysis, improving tumor staging, supporting treatment planning, and, ultimately, contributing to the main clinical decisions [1,2]. Such tools may incorporate deep neural networks for image analysis, within a complete end-to-end clinical pipeline, providing automatic image segmentation, image synthesis, and image-to-image translation [3,4,5]. Especially, in radiation therapy, image-to-image translation has attracted clinical interest for the potential of using cone-beam computed tomography (CBCT), instead of the conventional computed tomography (CT), for both diagnostics and planning purposes [6,7,8,9]. CBCT is traditionally used for imaging the patient in-room and registering intra- and inter-fractional changes in the target anatomy along the treatment course [1]. Whether the actual morphological changes significantly alter the anatomy on which the dosimetric distribution computed at the planning stage was based, a new CT scan is acquired to perform a planning update. Despite allowing for much faster imaging than CT fan-beam imaging, the CBCT technique introduces a significant amount of scattered radiations, resulting in image artifacts such as shading, cupping, reduced image contrast, and beam-hardening [10,11]. Moreover, the pixel values in CBCT images may fluctuate because of these artifacts, making them not directly usable for dose calculation unless some correction methods are applied to calibrate CBCT to CT scanner Hounsfield Unit (HU) values [12,13,14]. Thus, the availability of reliable techniques for the correction of CBCT images would extend their clinical usage, such as tumor shrinkage and organ shift evaluation, and treatment planning purposes, such as soft tissue segmentation for dose delivery computation. In addition, CBCT modality is compatible from a patient safety point of view (less additional non-therapeutic dose compared to traditional CT) with fractional dose delivery [1]. In this paper, generative deep neural networks were investigated for the translation of CBCT volumes into synthetic CT featuring compatible HU values. The same generative architecture was trained using supervised and unsupervised training techniques, exploiting a pelvic CT/CBCT publicly available dataset [15]. The comparison between the two obtained solutions was first quantified in terms of synthetic CBCT to original CT similarity by means of peak signal-to-noise ratio (PSNR) and structural similarity index measure (SSIM). The HU agreement was then evaluated in terms of mean absolute error (MAE).

### 1.1. Background

Correction and translation methods for quantitative CBCT can be broadly divided into hardware-based, prior-CT-based, and model-based approaches. Hardware-based methods remove scatter based on additional equipment mounted on the acquisition system. In particular, an anti-scatter grid and partial beam blockers, along with software estimation of scattering, was proposed for scatter suppression [16,17,18,19,20]. While this kind of approach has shown promising results, it is not always feasible to be implemented in clinical practice due to the installation and setup of additional devices on acquisition systems. Moreover, this type of solution inherently reduces the system quantum efficiency and can degrade image quality [21]. Prior-CT-based methods leverage information obtained from high-resolution planning CT applied to up-to-date information contained in CBCT. They can be based on processing techniques like deformable registration of CT to CBCT [22], or by linearly scaling the CBCT HU values to CT ones through histogram matching [23,24]. The shading artifacts inside CBCT can also be estimated by low-pass filtering the difference between the high-resolution CT projections and the raw CBCT ones and then subtracted from the CBCT during reconstruction [25]. In general, these methods achieved a limited solution to CBCT problems since they depend on the accuracy of the spatial alignment between CBCT and planning CT volume pairs from the same patients. This alignment is not always possible, particularly in the case of acquisition made on different days or in the presence of soft tissues. Model-based methods typically use Monte Carlo simulations to model the scatter contribution to CBCT projection data. Many researchers attempt to model the scatter distribution via some combination of analytical methods or Monte Carlo algorithms [26,27,28,29]. Model-based methods demonstrated to reproduce HU to sufficiently robust accuracy for clinical application because they rigorously simulate photon transport. However, their primary limitation comes from the physics models. Moreover, these techniques are limited by their execution time, usually in the order of minutes to hours, making them incompatible with an online application (e.g., dose evaluation or pre-treatment adaptation). In recent years, deep learning-based methods joined traditional approaches to improve the quality of medical images. These methods leverage the advance in the field of image-to-image translation, making use of convolutional neural networks (CNNs) to learn how to map voxels from a source distribution (e.g., CBCT) to a target one (e.g., planning CT). Rather than truly correcting scattering or other physical effects, CNN learns this type of nonlinear mapping iteratively. Therefore, this approach does not need complex physical model formulation required by, e.g., Monte Carlo-based methods. Moreover, although deep learning-based methods can be computationally expensive to train, once a well-trained model is developed, the image correction can be applied in seconds, making these methods compatible with the clinical practice. The literature presents two main paradigms to train CNN for image-to-image translation: supervised and unsupervised training. Supervised training requires paired images from two dominions (e.g., CBCT and CT) for model training. The input CBCT is processed by the model generating a synthetic CT (sCT), which is then compared to the corresponding ground-truth CT to minimize their pixel-by-pixel difference iteratively. Thanks to its neural encoding-decoding capabilities, the U-Net architecture is the most common one for this type of training [30,31,32,33]. Similar to prior-CT-based methods, images must be well-registered to achieve good performance when using supervised training. Since model training is based on a single pixel-wise loss function, the network could be biased and will tend to learn incorrect mapping in case of bad registration. It is not always possible to achieve such a good accuracy, especially in soft tissues, due to the patient’s weight loss, tumor changes, or the presence of air. On the other hand, unsupervised training enabled the possibility to use unpaired data for image-to-image translation [34]. The most common architecture for this kind of training is cycle Generative Adversarial Network (cGAN) [35]. This architecture is based on two concurrent subnetworks, a generator and a discriminator, which work in opposition. Given two different datasets, the generator tries to learn the mapping to convert one dataset to the other. The generator aims to trick the discriminator. On the other hand, the discriminator objective is to distinguish between real and synthetic images. This generator-discriminator cycle-consistent loop is designed to improve the generator ability to produce synthetic images that are virtually indistinguishable from real ones. Cycle GAN was first proposed for natural image synthesis. Still, recently various researchers demonstrated its application for many medical image synthesis tasks, like the generation of synthetic CT from CBCT [36,37,38,39], MR synthesis from CT images [40,41] or PET attenuation correction [42]. The main advantage of this method consists of the possibility to use unpaired, even unbalanced datasets, as the one-to-one correspondence between both dominions is no longer necessary. However, the computational power needed to train this architecture increases dramatically, since there are four models to be trained based on at least four distinct loss functions.

### 1.2. Key Contributions of the Work

In summary, the traditional approaches presented some limitations. Hardware-based methods require the installation and the setup of additional devices on acquisition systems. These devices involve additional costs and do not always guarantee high flexibility and generalization capability. For what concerns prior-CT-based methods, their most significant limitation is the dependence on planning acquisitions and the accuracy of spatial alignment with CBCT. This condition is not always easily obtainable, especially when the acquisition period between the two modalities increase. Monte-Carlo-based methods are the most accurate approaches, but they suffer from two main limitations. Their execution time is usually incompatible with in-room clinical application, and they have the critical requirements of a complex physical model formulation. On the other hand, the deep-learning-based methods proposed in this work aim to overcome some of these limitations. At first, it does not have any additional hardware requirements, making it of potential utility for every kind of CBCT acquisition system, e.g., robotic C-arm, gantry mounted systems or couch-mounted ones. Moreover, while training a deep convolutional neural network is a time-demanding task, they are very fast in generating output images in a production application once trained. The execution time for these methods is compatible with clinical use. Furthermore, the black-box nature of neural networks does not rely in general on any complex physical modeling. In order to obtain a network with sufficient generalization capability, it is mandatory to train it with a sufficiently broad and generalized dataset. The first generative model was based on the U-Net architecture trained according to a traditional supervised pattern. The second generative model was still a U-Net-based model, which was coupled to a discriminator network and trained without supervision in a cGAN configuration. Overall, two main impacts qualify the paper: (a) the feasibility of CNN to generate accurate synthetic CT scans from CBCT images, which is fast and easy to use compared to traditional techniques applied in clinics; (b) the proposal of guidelines to drive the selection of the better training technique, which can be shifted to more general image-to-image translation.

## 2. Materials and Methods

### 2.1. Dataset Description

The publicly available dataset, called Pelvic Reference Dataset (PRD) [15], used in this work was obtained from the Cancer Imaging Archive (https://www.cancerimagingarchive.net accessed on 10 February 2021). PRD included paired CT and CBCT volumes of the pelvic region of 58 subjects, spanning 28 males and 30 females. CBCT images were acquired at least 1 week after the corresponding CT volume acquisition. Moreover, PRD includes 47 and 11 volumes acquired in supine and prone positions, respectively. Two subjects (both female, one prone, and one supine) were removed from the dataset. In the first case, a metal hip prosthesis made it impossible to distinguish the anatomical structures in both CT and CBCT. In the second case, the CBCT field of view was too small to contain the entire subject’s pelvic region. Therefore, a massive part of the subject present in the CT was missing in its corresponding CBCT. These two cases were considered outliers and removed from the dataset, so that 56 cases were available for this study. Both CT and CBCT volumes featured size of 512×512 pixel on the axial plane with a pixel size of 1.00×1.00 mm and a single slice thickness of 3.00 mm. The number of slices of each CBCT was 88, while in CT scans that number was variable. Along with volume files, the PRD dataset also provided a table with the (x,y,z) coordinates to apply to the CBCT isocenter to align with their corresponding CT. To sum up, 4053 CT/CBCT 2D axial projections pairs were available. The entire dataset was split into train, validation, and test sets with a ratio of 80, 10, 10%, respectively, corresponding to 3243, 405, and 405 images per set.

### 2.2. Image Pre-Processing

Pre-processing steps were performed before feeding data to the neural network. At first, a binary mask was created to separate the subject from any non-anatomical content (e.g., treatment couch). This masking procedure avoids any negative impact from these structures on training procedures. In order to prepare these masks, Otsu thresholding was applied to each volume. A max connected component analysis was then performed on each mask to remove any non-anatomical residual finely. A final binary fill holes filtering and erosion filtering were applied to enhance mask accuracy further. Finally, the mask was applied to the original volume, obtaining an isolated anatomical region. Corresponding masked CT and CBCT were then rigidly registered, using isocenter coordinates provided with PRD Dataset. This step was necessary only for supervised training since the unsupervised one did not need paired data. However, good alignment was also required for evaluation analysis. The HU range of the grayscale values was first clipped to [−1024,3200] and then rescaled to [0,1] with a linear mapping. In order to reduce the computational cost for CNN, every axial slice was resampled to 256×256 pixel.

### 2.3. Deep Convolutional Neural Network Models

The two generative models were both based on the U-Net architecture trained according to two different training patterns. The supervised generative model required only this network, while the unsupervised one also required the definition of a discriminative architecture. In order to make the comparison possible, the generator architecture was precisely the same for both supervised and unsupervised procedures. The generator was implemented as an adapted version of the popular U-Net architecture. The basic U-Net is primarily used to solve pixel-by-pixel classification problems in image segmentation [43]. In the present work, U-Net was adapted to solve an image-to-image conversion task to generate sCT images from CBCT. The basic building blocks for the generator were depicted in Figure 1. The first one, called the ConvBlock, was based on a modified version of the implementation of Isola et al. [34]. It was composed of a 2D convolution with a 3×3 kernel, followed by an instance normalization layer and a swish activation function. Instance normalization was demonstrated to improve the performance in image generation tasks [44]. ConvBlock was present in the network both standalone and as a part of another processing block, called the InceptionBlock. This latter block was adapted from GoogLeNet [45]. It comprised four parallel ConvBlocks, each with an increasing kernel size of dimensions 1×1, 5×5, 7×7, and 11×11. In this way, the input received by the InceptionBlock was processed simultaneously by multiple receptive fields. The output of each branch was then concatenated, and the entire stack of feature maps was returned as output. The main purpose of this processing block was to perform multi-scale feature extraction from the input image. These multi-scale extracted features, ranging from small to large receptive fields, can provide better results for image synthesis.

The overall generator structure, depicted in Figure 2, was composed of a contracting and an expanding path, both based on the basic processing blocks. The first two upper processing blocks of the generator were composed of InceptionBlocks, while the deeper three were composed of ConvBlocks. In this way, the network was divided into two parts with two different functions: the inception part extracted global contextual information while the traditional part has the task to capture finer context and precise localization.

The discriminator (Figure 3) was a CNN that performs image classification. Its architecture was based on the PatchGAN architecture [34], considered to be the gold standard discriminator for cGAN [35,38,39]. It consisted of four consecutive ConvBlocks, with a 4 × 4 kernel size. The convolution for the first three ConvBlock was set with stride 2, giving as outputs a tensor with half the size and twice the features map. The last ConvBlock had stride one and maintained the size and the number of feature maps unchanged. A sigmoid activation function followed the last layer, producing a 32 × 32 map with every pixel in the [0,1] range. This output map was used for patch-wise classification of the input image as real or fake.

### 2.4. Training Methods

The two training routines are schematized in Figure 4. For supervised training, given the original CBCT image as input to generator CT (G${}_{CT}$) network, the generated sCT is compared to the corresponding ground-truth CT. During training, the network adapts its weight according to MAE loss function, expressed as:(1)$$L_{supervised}\left(CT,sCT\right)=\frac{1}{N}\sum_{i=0}^{N}|CT_{i}-sCT_{i}|$$

In regard to the unsupervised learning, the cGAN structure incorporated two generators and two discriminators, competing against one another, namely generator CT (G${}_{CT}$), generator CBCT (G${}_{CBCT}$), discriminator CT (D${}_{CT}$), and discriminator CBCT (D${}_{CBCT}$). G${}_{CT}$ was used to generate sCT from CBCT, while G${}_{CBCT}$ was used to generate synthetic CBCT (sCBCT) from CT. On the other hand, D${}_{CT}$ was used to distinguish between real CT and sCT, and D${}_{CBCT}$ is used to distinguish real CBCT from sCBCT. In the first step of the training, G${}_{CT}$ (G${}_{CBCT}$) took CBCT (CT) as input and generated sCT (sCBCT). Then, G${}_{CT}$ (G${}_{CBCT}$) took sCT (sCBCT) as input and generated a cycleCBCT (cycleCT), which is supposed to be equal to the original CBCT (CT). Meantime, D${}_{CT}$ (D${}_{CBCT}$) tried to discriminate between real CT (CBCT), labeled as 1, and sCT (sCBCT), labeled as 0. Generator loss functions included three types of terms: adversarial loss, cycle consistency loss, and identity loss. Discriminator loss was composed only of an adversarial term. These loss functions were combined to mapping the distribution of the generated images to the distribution domain of the target images (see Appendix A).

### 2.5. Performance Metrics

In order to quantitatively evaluate network performances, three widely accepted numerical metrics were used: peak signal-to-noise ratio (PSNR), structural similarity index measure (SSIM), and mean absolute error (MAE). PSNR is computed as the ratio between the maximum possible power of a signal and the mean square error of the images being compared. Its value is measured in decibel and approaches infinity as the mean squared error between sCT and ground-truth CT approaches zero. Therefore, a higher PSNR value corresponded to higher image quality and vice-versa [46]. SSIM was first introduced in 2004 by Wang et al. [47]. It measures similarity between two images based on three factors: luminance, contrast, and structure. Compared to PSNR, SSIM provides an indication of similarity closest to that of the human visual system. The value for this metric ranges between 0 and 1, where 1 indicates the best possible level. Lastly, MAE is used to evaluate the HU accuracy between two images quantitatively. Before computing its value, the amplitude of the network output, i.e., [0,1], was scaled back to the original range, i.e., [−1024, 3200]. The lower the MAE, the higher the HU accuracy of the two images. Every metrics was evaluated using CT as the ground truth reference.

### 2.6. Cross-Validation Analysis

In order to validate the performance of the trained models, a four-fold cross-validation experiment was carried out on both methodologies. The entire dataset was divided into four subsets, each subgroup consisting of images of 14 patients. For each experiment, three subsets were used as the training set, while the remaining one was used as the test set. The cross-validation allowed a comparison of the performance of the models against the baseline results, i.e., the difference between the CT and the original CBCT images. The improvement in terms of SSIM, PSNR, and MAE was quantitatively analyzed. The statistical difference between the candidate models was also evaluated with Kruskal-Wallis non-parametric test for median differences (*p* < 0.01) and Tukey–Kramer post-hoc comparison.

### 2.7. Implementation Details

The Keras [48] and TensorFlow [49] Python frameworks were used to develop the network models, loss functions, metrics, and training routines. The training was carried out in the Google Colaboratory Cuda-enabled environment, equipped with a four-core CPU, 25 GB RAM, and NVIDIA^®^ Tesla^®^ P100 GPU support 16 GB RAM. The number of epochs has been set to 25. When the validation set SSIM score is maximized, the training routine was configured to save the set of the best network weights. The training was optimized with ADAM (Adaptive Moment Estimation) optimizer [50], with the following parameters: learning rate $2\times 10^{-4}$, exponential decay rate for the first moment estimates β1=0.5, and exponential decay rate for the second-moment estimates β2=0.999. The batch size was set at 10. In order to generalize network performance as much as possible and prevent overfitting, data augmentation was performed at run-time during training. At the beginning of each training epoch, just before giving the image as input to the network, a series of image elaborations were applied. In particular, the images were randomly rotated by multiple of 90 degrees and horizontally flipped. The same transformations were also applied to the corresponding ground truth CT in order to maintain coherence between pairs. By applying these transformations, each CBCT/CT pair featured eight different configurations (four rotations times two flips). This implied that at each iteration the network was fed with different images. As an example, assuming that we have *N* images in the training dataset, at each iteration step of the training *N* different images are generated from the original samples applying the described random transformations.

## 3. Results

Supervised training required about 3 minutes per epoch. The only architecture used for this training pattern was the generator, corresponding to 2,554,977 trainable parameters. On the other hand, unsupervised training required about 5 minutes per epoch due to its increased complexity. This method featured two identical generators and two identical discriminators. Being the discriminator network characterized by 429,185 trainable parameters, the overall number of weights for the unsupervised model was 5,968,324. For what concern inference time, the two approaches were comparable. Generating sCT required less than 4 seconds for an entire CBCT volume (~70 slices) in both cases, computed on the same GPU environment used for training.

### 3.1. Performance Metrics

The cross-validation experiments confirmed that the proposed CNN models improved the metrics with respect to the baseline (Figure 5), being median and interquartile results summarized in Table 1. In general, supervised training attained better results for every metric. Considering SSIM, supervised sCT obtained an improvement of 2.5% with respect to the baseline (*p* < 0.0001), while unsupervised sCT gained an improvement of 1.1%. PSNR also confirmed this trend, resulting in an enhancement from the baseline for supervised sCT and unsupervised sCT of 4.19 dB (*p* < 0.0001) and 2.3 dB (*p* < 0.0001), respectively. The relative gains for PSNR were 15.7% and 8.6%. Lastly, the supervised model reduced the mean absolute error between supervised sCT and CT by 58.16 HU (62.3%), while the unsupervised model resulted in a reduction of 46.92 HU (50.3%). Even in this case, the supervised model appeared significantly better than the unsupervised one (*p* < 0.0001).

### 3.2. Qualitative Comparison

Some examples of sCT generated by the supervised and unsupervised approaches are depicted in Figure 6. The first column represents the ground truth CT image, while the second one contains the original CBCT input. The third and fourth columns show the generated sCT images predicted from supervised and unsupervised models, respectively. The last column compares one intensity profile (the central row depicted with a line) for every imaging modality. The same example slices are also showed as difference maps in Figure 7. Every column represents the difference between the modality under evaluation (CBCT, supervised sCT, and unsupervised sCT) and the ground truth CT.

The first row represents the case of a CBCT/CT pair with a similar field of view (FOV). From the intensity profile and the difference map, it can be noticed that both models enhanced overall image qualities in terms of HU mapping, with slightly better visual details for the supervised model. The second row shows an example in which CT FOV is wider than CBCT one. In this case, it can be observed that the supervised model tried to compensate for the missing structures at the border with unrealistic values. Contours of supervised sCT were blurred and unreliable, while unsupervised sCT coped better with the original CBCT boundaries. In more detail, the unsupervised model did not attempt to recreate the missing FOV, resulting in more realistic and reliable contours with respect to the original CBCT. Concerning the internal anatomical structures, both models reached good qualitative results, as demonstrated by the corresponding intensity profile and difference map. The last row in Figure 6 and Figure 7 shows a case in which the rectum area had an air-filled cavity with different shapes between the CT and the CBCT. The supervised model reacted by filling the air cavity with the surrounding values, while the unsupervised model preserved its contours. This latter behavior is preferable, as it is assumed that the information content of the CBCT is more updated than that of the CT. Again, the supervised model attempted to scale the HU values without considering possible changes in the anatomical structures of interest.

## 4. Discussion

### 4.1. Main Findings

This work proposed a deep-learning-based approach for generating synthetic CT images from CBCT scans, featuring HU values compatible with the traditional CT domain, providing a fair comparison between supervised and unsupervised training paradigms. As demonstrated by the cross-validation analysis, the supervised model outperformed the unsupervised ones. The supervised network guaranteed advantages in terms of computational cost, having 57% fewer weights to train and requiring 40% less training time for the same number of epochs. Moreover, it gained an increase of 12% in terms of MAE with respect to the unsupervised one, showing a superior ability in mapping HU values. On the other hand, the supervised model showed to be not always reliable, resulting in some unexpected behavior, especially in the contour regions. This kind of artifact did not appear in images generated through the unsupervised model, which proved to be more reliable in preserving the anatomical structure and the up-to-date information contained in the CBCT scan. One example of this is clearly visible in the bottom row of Figure 6, where the shape of the air cavity present in CBCT is preserved only for the unsupervised case, while the others tended to close the air gap. The unsupervised model did not suffer from this condition due to the cyclic loss functions, consisting of three main terms, each with a different purpose. The more straightforward loss function defined for the supervised model, i.e., mean square error, specialized the network to learn mapping without adequately considering the anatomical structure.

### 4.2. Comparison with the Literature

To our best knowledge, just one comparison between supervised and unsupervised training of deep networks, addressing back and forth translation between CT and magnetic resonance images was proposed in the recent literature [41]. In such a work, the authors explored different network architectures proposing a modified U-Net (supervised training) and the ResNet model for the generator (unsupervised training) proposed in [35]. Conversely, in our work, the same model was adopted in both the supervised and unsupervised generative parts, which smoothed out any biases in the comparison. Approaches competitive with our work were reported in the recent literature. Using an MC-based methodology, HU correction of about 31% for five lung cancer patient images was gained [13]. Another proposal, which used histogram matching on ten prostate cancer patient scans, resulted in a 20% HU correction [24]. A phantom-based study was carried out on the Catphan 600 (The Phantom Laboratory, Salem, NY, USA) [25] achieved an overall accuracy of 95% in HU recovery. Nonetheless, results on patient images were not presented. Concerning deep-learning-based methods, a proposal based on the U-Net trained with a supervised approach on 30 prostate cancer patients presented an HU accuracy enhancement of about 69% [30]. A comparative study explored the performances of some supervised models trained on 42 prostate cancer patients. The results obtained by the three tested models were 55, 17, 47% in terms of HU relative improvement [33]. Exploring the use of cGAN and unsupervised training, a study disclosed an enhancement of 57% with a dataset containing 120 head and neck scans [36]. Another unsupervised-based research resulted in a 16% increase for 33 prostate cancer patients [37]. The results proposed in this work are generally in line with the prior art, with an HU improvement of 62% and 50% for the supervised and unsupervised models, respectively.

### 4.3. Technical Challenges and Work Limitations

The use of complex models for remapping HU can be considered oversized when looking at the intensity profiles (cfr. Figure 6), where linear shifts appear predominant. This would suggest that regression models may be enough to address HU mapping. However, the inter-variability between the two modalities (e.g., air gaps, morphological variations due to postures, lesion progression) may easily introduce non-linearity in the intensity profile mapping (cfr. Figure 8). In the upper panel of Figure 8, an air gap can be appreciated in the CBCT and in the corresponding CT. The notch in the intensity profiles is just produced by such a gap. This kind of deformation is difficult to correct by simple linear mapping. In the lower panel, relevant beam hardening of the CBCT at the image boundaries, corresponding to peripheral density appearance, is traceable in the corresponding intensity profile (orange curve) as an undue convexity at the borders. Likewise, a linear transform of the profile does not ensure accurate correction. Consequently, convolutional neural networks represent a natural approach to address as they feature input/output non-linear transforms and good generalization capabilities.

As we have shown, quantifying the image translation effectiveness by means of SSIM and PSNR can be inconclusive. For example, both lower SSIM and PSNR values (supervised against unsupervised) do not necessarily indicate that the images are qualitatively worse (cfr. Table 1). In particular, SSIM is a perception-based metric considered to be correlated with the quality perception of the human visual system, and the higher its value is, the better is the perception of similarity for human eyes [47]. Still, some studies have shown that SSIM is less effective when used to evaluate blurry and noisy images [51,52]. This reduces the relevance of comparisons performed using such metrics. Conversely, MAE provided an objective quantification of the HU matching. As far as the selection between the two approaches is concerned, we may argue that the supervised approach is preferable in the presence of a large amount of paired data, ensuring high accuracy in HU mapping. Nonetheless, gaining this advantage requires that CBCT and CT images be gathered as close as possible to minimize the anatomical difference. It is also crucial to mask images according to the smallest FOV between the two modalities in order to have comparable information content in between. In detail, masking ensures similar FOVs preventing the supervised network from attempting to reconstruct the sCT in the CBCT regions where this information is missing. In order to reduce the dependency on such constraints, the unsupervised method is suitable, avoiding pairing, requiring fewer pre-processing steps (e.g., masking) and being more conservative in terms of anatomical features. Removing the constraint on the pairing, the training effectiveness can benefit from a large amount of data, easier to collect. Nonetheless, even though obtaining unpaired images is clinically simpler, using explicitly paired data in training reduces geometric discrepancies at body boundaries. As verified in [21], unsupervised training may take advantage of paired data allowing to better focus on soft-tissue contrast than geometric mismatches. To further validate the methods presented in this work, a dosimetric analysis would be required. However, the chosen dataset did not allow this type of study to be explored due to the lack of the original CT dose distribution plan. Moreover, our methodologies focused only on a single anatomic district, the pelvic one. We wanted to demonstrate the feasibility of both methods in a body region particularly subjected to inter- and intra-fractional changes. The comparison between the two training strategies was made to show their different approach in addressing image-to-image translation tasks, highlighting their strengths and limits.

## 5. Conclusions

We have shown that the proposed methodologies allow generating accurate synthetic CT images by automatically processing CBCT scans, featuring processing time faster than traditional Monte-Carlo-based methods. The basic idea behind this approach is to correct CBCT images exploiting their potential not only in a more accurate patient positioning, but also for clinical evaluation purposes as tumor shrinkage and organ shift. For what concerns the integration in the clinical workflow of treatment re-planning, this method can be integrated into a tool to be deployed that elaborates the reconstructed CBCT images, acting as an image filtering process. The correction of CBCT also makes possible the direct morphological comparison with the planning CT. As a consequence, the dose planning update, according to the new patient condition following fractional treatment, is just one of the possible usages of the CBCT, corrected with realistic tissue density according to HU. The clear advantage of using the in-room calibrated CBCT images for online re-planning consists of the availability of the actual patient anatomy. Nonetheless, this is still a challenge in radiation therapy and the subject of extensive research, which we plan to investigate in future works.

## Figures and Tables

**Figure 1 diagnostics-11-01435-f001:**
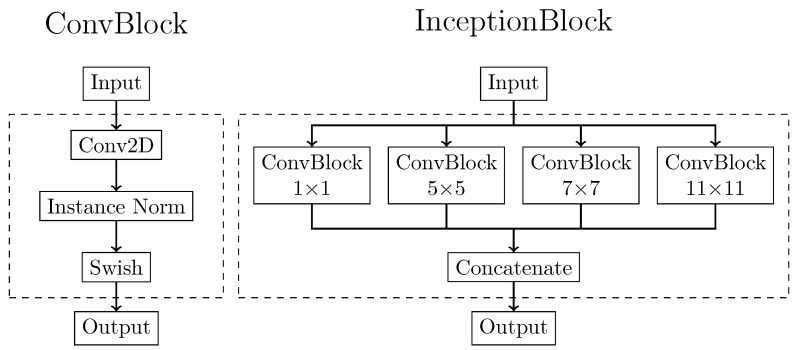
Basic building block architectures. The ConvBlock (**left**) is composed of a 2D convolutional step with kernel variable kernel size, followed by instance normalization and a Swish activation function. The InceptionBlock (**right**) is composed of the parallel combination of more ConvBblock with kernel dimension of 1×1, 5×5, 7×7, and 11×11. The output of each ConvBlock is then concatenated in a single output tensor.

**Figure 2 diagnostics-11-01435-f002:**
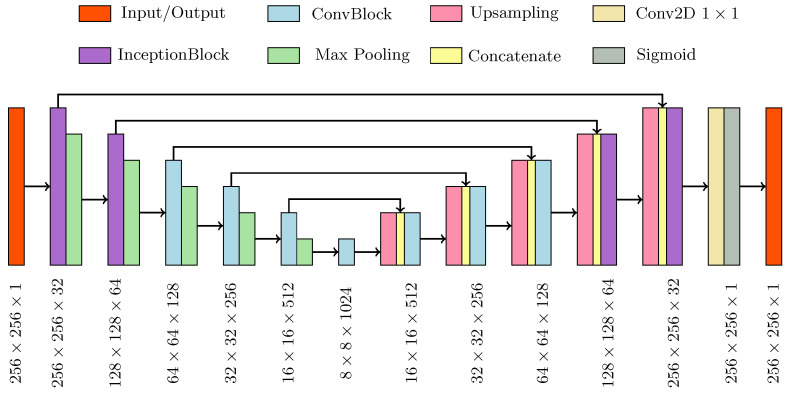
Schematic of the generator network architecture.

**Figure 3 diagnostics-11-01435-f003:**
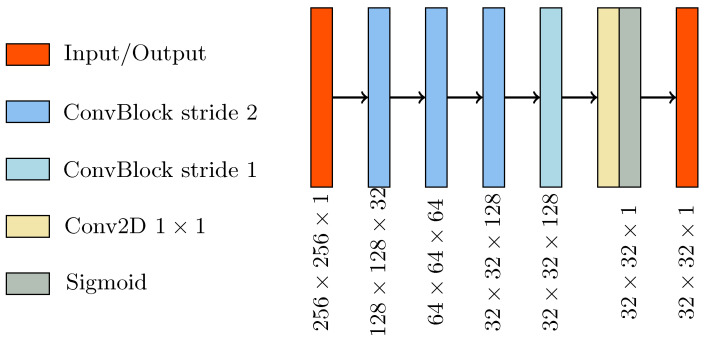
Schematic of the discriminator network architecture.

**Figure 4 diagnostics-11-01435-f004:**
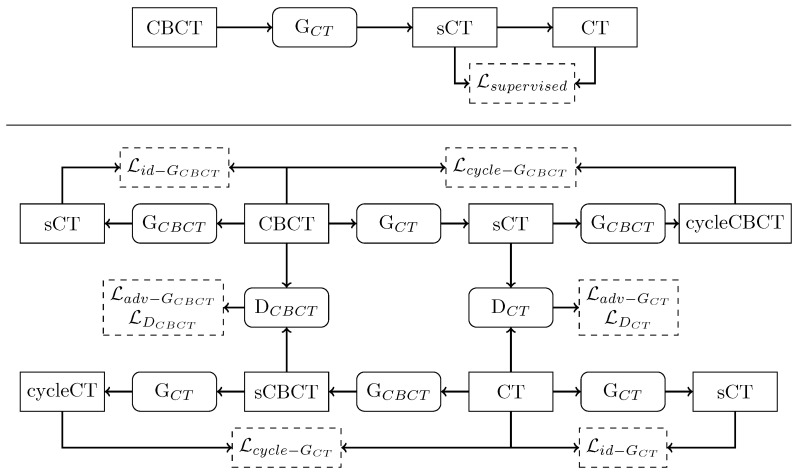
Schematic flow chart of supervised (**upper panel**) and unsupervised (**bottom panel**) training routines. Rectangle boxes represent images, and rounded corner boxes depict neural network models. Dashed boxes indicate a loss function. For a detailed explanation of the training routines, refer to the Appendix A.

**Figure 5 diagnostics-11-01435-f005:**
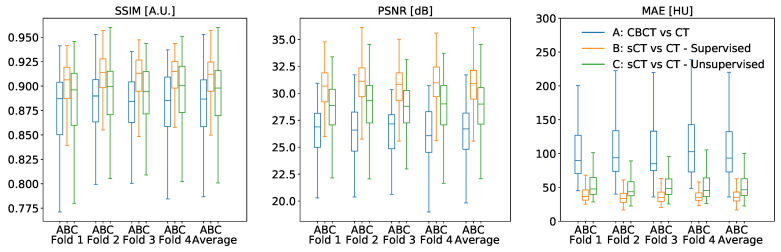
Quantitative analysis of SSIM, PSNR, and SSIM values between original CBCT, sCT supervised, and sCT unsupervised against the corresponding CT, computed for each fold of the four-fold cross-validation.

**Figure 6 diagnostics-11-01435-f006:**
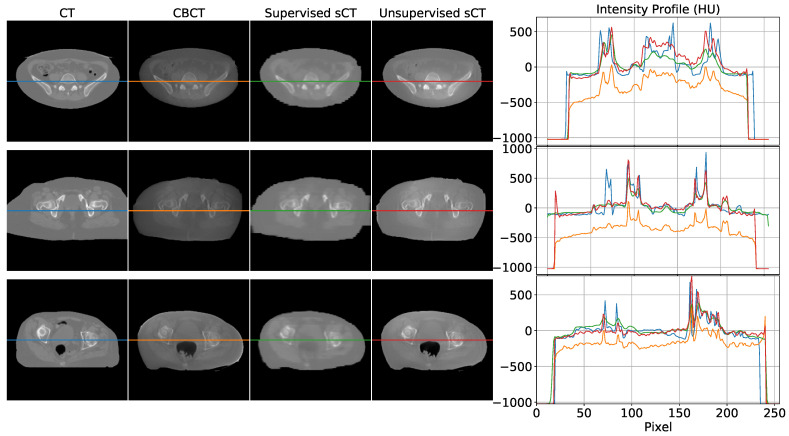
Visual comparison between some axial slices depicted in every modality: CT, CBCT, sCT supervised, and sCT unsupervised. Every row corresponds to a different example taken from a different subject. The rightmost part of the figure compares the intensity profiles of the central line of the images, highlighted by the central line in the four representations. Images are displayed with Window = 2000, Level = 0.

**Figure 7 diagnostics-11-01435-f007:**
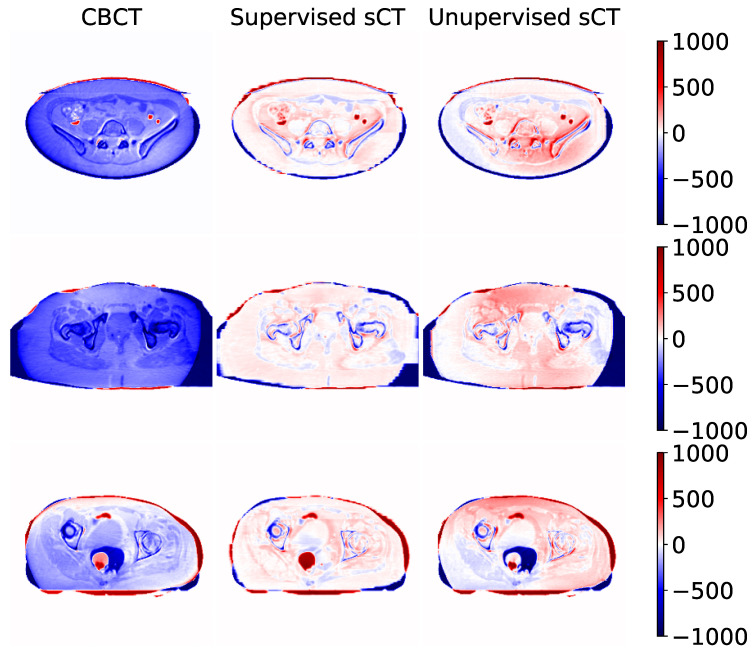
Difference maps computed between original CBCT, sCT supervised, and sCT unsupervised using their corresponding CT as reference. Numeric values are in Hounsfield Units. Each row represents a different example taken from a different subject. The examples depicted in this figure correspond to the example presented in Figure 6.

**Figure 8 diagnostics-11-01435-f008:**
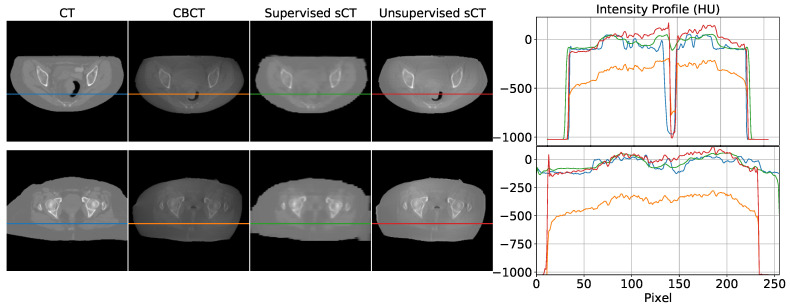
Examples of non-linearity of CBCT scans. Air gap introduced a notch in the intensity profile (**top**). Beam hardening is visible as a convex shape in the orange curve of both examples (**bottom**). Images are displayed with Window = 2000, Level = 0.

**Table 1 diagnostics-11-01435-t001:** Performance metrics evaluated on original CBCT, sCT supervised, and sCT unsupervised. Every value is computed against the original CT, considered as the ground truth. Every value is expressed as median (interquartile range).

	SSIM [A.U.]	PSNR [dB]	MAE [HU]
CBCT	0.887 (0.048)	26.70 (3.36)	93.30 (59.60)
Supervised sCT	0.912 (0.030)	30.89 (2.66)	35.14 (13.19)
Unsupervised sCT	0.898 (0.046)	29.00 (3.38)	46.38 (24.86)

## Data Availability

The data presented in this study are openly available in The Cancer Imaging Archive (TCIA) at 10.7937/TCIA.2019.woskq5oo, accessed on 23 June 2021. These collections are freely available to browse, download, and use for commercial, scientific and educational purposes as outlined in the Creative Commons Attribution 3.0 Unported License.

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
