# Peer review of "Comparison of Supervised and Unsupervised Approaches for the Generation of Synthetic CT from Cone-Beam CT"

_diagnostics, 2021, doi:10.3390/diagnostics11081435_

Round 1
Reviewer 1 Report
The proposed methodology is complex and difficult to implement on a large scale.
The advantage of CBCT over CT appears unclear, particularly in terms of delivered dose.
Reviewer 2 Report
The article reports the implementation of a supervised and unsupervised approaches for the generation of synthetic CT from cone-beam CT and an extensive comparison between them.
1) in 2.1, the number of images included in the dataset and used for the study should be given; otherwise the reader finds this information out only when the splits are mentioned.
2) in 2.7, the augmentation is mentioned marginally. However, I think it deserves more detail. How many images does the augmentation step create during training?
3) Regarding the comparison with work 41, I would add some results obtained by them, to make the comparison with your work more immediate.
4) For me, the manuscript lacks a discussion (even a brief one) about what the implications for treatment might be with their proposal and how their results can be integrated into clinical practice.
5) Furthermore, the conclusions are very brief. Although the rest of the manuscript is well written and motivated, I think the authors should expand the conclusion section to make a point about the results obtained and the discussions presented throughout the paper, in a concise manner. Some of the suggestions from point 3) can also be added here, as a development of future work, for example.
To sum up, the manuscript is well written and extremely well organised. I really appreciated the discussions made by the authors and the comparison with the state of the art. For the latter, I would have suggested a table to report the work, but I think this is no longer necessary as there is only one work with which the authors can make a direct and fair comparison. In fact, the others are just examples of similar applications in different scenarios.
In general, the main quality of this work is the extensive descriptions and discussions in each part. I also appreciated how the authors described that "SSIM and PSNR values do not necessarily indicate that the images are qualitatively worse", which is a thing often not taken into account in this field.
In conclusion, I think the manuscript is good for publication, once the authors have addressed the minor suggestions provided.
Round 2
Reviewer 1 Report
Nothing